# Nondestructive Wafer Level MEMS Piezoelectric Device Thickness Detection

**DOI:** 10.3390/mi13111916

**Published:** 2022-11-05

**Authors:** Yongxin Zhou, Yuandong Gu, Songsong Zhang

**Affiliations:** 1School of Microelectronics, Shanghai University, Shanghai 200444, China; 2Shanghai Industrial μTechnology Research Institute, Shanghai 201899, China

**Keywords:** picosecond ultrasound, nondestructive thickness measurement, MEMS piezoelectric device/sensor, scandium doped aluminum nitride (AlScN)

## Abstract

This paper introduces a novel nondestructive wafer scale thin film thickness measurement method by detecting the reflected picosecond ultrasonic wave transmitting between different interfacial layers. Unlike other traditional approaches used for thickness inspection, this method is highly efficient in wafer scale, and even works for opaque material. As a demonstration, we took scandium doped aluminum nitride (AlScN) thin film and related piezoelectric stacking layers (e.g. Molybedenum/AlScN/Molybdenum) as the case study to explain the advantages of this approach. In our experiments, a laser with a wavelength of 515 nm was used to first measure the thickness of (1) a single Molybdenum (Mo) electrode layer in the range of 100–300 nm, and (2) a single AlScN piezoelectric layer in the range of 600–1000 nm. Then, (3) the combined stacking layers were measured. Finally, (4) the thickness of a standard piezoelectric composite structure (Mo/AlScN/Mo) was characterized based on the conclusions and derivation extracted from the aforementioned sets of experiments. This type of standard piezoelectric composite has been widely adopted in a variety of Micro-electromechanical systems (MEMS) devices such as the Piezoelectric Micromachined Ultrasonic Transducer (PMUT), the Film Bulk Acoustic Resonator (FBAR), the Surface Acoustic Wave (SAW) and more. A comparison between measurement data from both in-line and off-line (using Scanning Electron Microscope) methods was conducted. The result from such in situ 8-inch wafer scale measurements was in a good agreement with the SEM data.

## 1. Introduction

Piezoelectric materials convert electrical energy into mechanical energy, or vice versa. Due to their unique properties, piezoelectric materials have been used in many applications, which include infrared (IR) detectors [1], healthcare and sports monitoring [2,3], RF Filters [4,5,6], viscosity sensors [7], etc. Piezoelectric devices are usually made of stacked multilayer thin films. Taking traditional FBAR as an example, the overall piezoelectric stack is a sandwiched structure of two electrode layers and one piezoelectric layer [8]. The exact thickness of each layer has an impact on the overall performance of the device. For example, in FBAR as a temperature sensor, the device sensitivity changes from 546 kHz °C^−1^ to 190 kHz °C^−1^ as the insertion (Ti) thickness changes from 20 nm to 50 nm [9]. In FBAR for a high-frequency resonator, the thickness of the piezoelectric layer (E.g. Aluminum Nitride) has an impact on frequency [8]. Thus, a precise sub-surface thin film thickness control (normally in sub nanometer scale) is critical to ensure good device performance. 

Due to its excellent accuracy and capacity for multiple layer detection, ellipsometry is frequently utilized in the detection of thin film thickness [10]. However, it struggles to function properly with opaque layers. Most of other popular sub-surface detection methods are based on the use of penetrating electron beams or waves to detect materials. For example, the famous Transmission Electron Microscope (TEM) and the High-Resolution Transmission Electron Microscope (HRTEM) realize the sub-surface imaging of materials through the cross-sectional analysis of atomic resolution. However, these detection methods are destructive, and not suitable for wafer level in-line process monitoring [11]. Instead of electromagnetic waves and electron beams, ultrasonic acoustic echo is also widely used for film crack detection and imaging in both material science and medical application. However, the resolution is usually limited to tens of microns in scale, which is not sufficient for thin film measurement [12]. This paper will introduce picosecond ultrasonic pulse detection technology, which overcomes the limitation of low detection resolution, and introduces no damage to thin film on the processing wafer. This technology has already been used to characterize Bulk Acoustic Wave (BAW) [13,14,15]. However, it has not been widely used. This paper will systematically study the AlScN piezoelectric devices that have been popular in the field of piezoelectric MEMS in recent years. We chose universal substrate (Si, SOI), material (AlScN, Mo), structure (typical sandwich structure), and finally successfully characterized the piezoelectric stack on SOI. This systematic study will be of great help to piezoelectric MEMS fabrication.

## 2. Theory of Picosecond Laser Ultrasonics 

### 2.1. Opaque Film

Picosecond laser ultrasound is a method for studying materials using high-frequency acoustic pulses generated and detected by ultra-short optical pulses (typical pulse duration < 1 ps) [16]. When such an optical pulse, known as a pump pulse, is incident on the surface of an opaque solid, some optical energy is absorbed and converted to heat. The production of heat in solids causes the lattice temperature to rise, and this leads to thermal stress that launches a strain pulse propagating in three dimensions. We focus on acoustic propagation to the surface, because its high frequency (usually in the range of 10–1000 GHz) leads to small wavelengths, resulting in high resolution in thin film thickness measurement [17,18,19]. When the acoustic pulse returns to the area irradiated by the probe pulse, the optical properties of the material are modified. This modification affects the reflectivity of the probe pulse. This process is illustrated in Figure 1 as the example of an opaque thin film on a substrate [20]. 

Typical opaque film reflectivity waveform is shown in Figure 2b. The peak shows the thickness of the opaque film.

### 2.2. Transparent Film

The measurement of opaque films is an acousto-optic interaction which is based on the modification of the optical properties of surface materials. However, this does not apply to measurement of transparent films. 

First, as mentioned above, the opaque film absorbs light energy, converts it into heat energy, and then converts it into acoustic pulses. However, transparent film cannot be used as a photo-acoustic conversion layer. 

The value of absorption of laser beam is related by:

Absorptivity = 1 − Reflectivity (for opaque materials)

Absorptivity = 1 − Reflectivity-Transmissivity (for transparent materials)

Transmissivity of film with thickness L is shown in Equation (1) [21].
(1)$$\begin{array}{c} T=\frac{{\left[1-\frac{{\left(n-1\right)}^{2}+k^{2}}{{\left(n+1\right)}^{2}+k^{2}}\right]}^{2}e^{\frac{-4\pi kL}{\lambda }}}{1-{\left[\frac{{\left(n-1\right)}^{2}+k^{2}}{{\left(n+1\right)}^{2}+k^{2}}\right]}^{2}e^{\frac{-8\pi kL}{\lambda }}} \end{array}$$
where *n* is the refractive index, *k* is the extinction coefficient, and *λ* is the laser wavelength.

The reflectivity is as follows:(2)$$\begin{array}{c} R=\frac{{\left(n-1\right)}^{2}+k^{2}}{{\left(n+1\right)}^{2}+k^{2}}+\frac{{\left(n-1\right)}^{2}+k^{2}}{{\left(n+1\right)}^{2}+k^{2}}Te^{\frac{-4\pi kL}{\lambda }} \end{array}$$

The *k* of the transparent material is close to 0, so the absorption rate is also close to 0 by Equations (1) and (2). Take the transparent material AlN as an example. When the laser wavelength is 515 nm, *n* is 2.1647, *k* is close to 0. We can calculate that if *T* is close to 0.761423, *R* is close to 0.238575, and absorptivity is close to 0.

In summary, transparent films absorb light energy very poorly, and convert very little thermal energy as a result.

Furthermore, as shown in Figure 2a, the effect of the acoustic pulse, unlike opaque film, only affects the reflected signal when it reaches the surface of the film. The acoustic pulse continues to have an effect throughout the transparent film.

When probing transparent materials, picosecond ultrasound technology is also known as picosecond acoustic interferometry. The probe laser pulse interacts with phonons that satisfy the momentum conservation law for photon-phonon, photo-elastic interaction, that is, they satisfy the Brillouin scattering condition [22].

From one perspective, acoustic phonon propagating through the material can produce periodic density fluctuations (permittivity fluctuations) as a moving grating. Brillouin scattering can be explained by the diffraction of the incident photons from this grating, which causes the frequency-shift of the scattered photons with a Doppler effect [23,24]. The measured signal varies with time because the relative phase of the light scattered by the acoustic pulse and reflected by immobile interfaces continuously changes with time due to the variation in the spatial position of the acoustic pulse. The acoustically induced oscillating contribution to the reflectivity signal (see Figure 2c) is commonly known as the Brillouin oscillation [25]. 

## 3. Experimental Details 

### 3.1. Experimental Setup

The picosecond ultrasound tool used in this experiment is the MetaPULSE G from Onto Innovation, Wilmington, NC, USA. Its basic function is is shown in Figure 3. The laser produces 0.2 fs optical pulses at a repetition rate of 60 MHz, and the wavelength (λ) is 515 nm. The laser output is split to provide pump and probe beams with crossed polarizations. The probe path length can be incrementally increased by moving a retro-reflector mounted to a translating delay stage. Delay time (Δt) between the pump and probe beams increases as the probe path length increases. Both beams are focused on the same point of the sample by a lens. Finally, the detector observes the change of reflectivity and obtains the time domain map of reflectivity. Another commendable point of this tool is that it has its own function to simulate thermal background noise, which is the biggest disturbance to reflectivity waveform judgment. The change in waveforms after thermal background subtraction, or lack of, is shown in Figure 3.

We observe 4 points (0,0) (30,0) (60,0) (90,0) of each sample by SEM. The 8-inch wafer will inevitably have uniformity problems, and taking these 4 points can essentially cover the full range of the 8-inch wafer’s thickness, which better illustrates the applicability of picosecond ultrasonic detection technology.

### 3.2. Description of the Sample

The ultimate purpose of this paper is to detect the standard piezoelectric composite structure (electrode layer/piezoelectric layer/electrode layer). The selected electrode layer sample is the most commonly used electrode material, Mo. The selected thickness is to be in the range of 0.1 μm to 0.3 μm; this range fits the process requirement in most practical cases. The piezoelectric material of choice is AlScN, which is widely used in acoustic MEMS devices due to its high acoustic velocity and excellent stability at high temperatures [26]. AlScN film with a thickness range of 0.6–1 μm is targeted to align with the fabrication needs for low frequency MEMS device. 

As shown in Table 1, eight samples were measured in this paper. We can divide them into three categories. (1) samples (A, B, C, D) are Mo and Al_0.9_Sc_0.1_N single-layer films of different thickness. (2) Sample E is a bi-layer film with Al_0.9_Sc_0.1_N on top and Mo on the bottom. In sample F, the piezoelectric layer is replaced with Al_0.8_Sc_0.2_N. Sample G is distinguished from sample E by the order of Mo and Al_0.9_Sc_0.1_N deposition. (3) Sample H is a typical piezoelectric composite structure. 

Scandium-rich abnormal grains are found on the surface of AlScN after sputter [27]. Large abnormal grains can lead to mistakes in the picosecond ultrasound detection. In this experiment, each sample was deposited on 8-inch Si substrates by reactive magnetron sputtering in a Sigma ® Deposition System from SPTS, Newport, Wales, UK. Low abnormal grain precipitation and tiny size on the AlScN surfaces are produced by this machine in conjunction with a competent process. As a result, abnormal grains had little effect on the experiment. Furthermore, AlScN is an insulator. It is easy to blur at the AlScN/Air interface when observed by SEM. Therefore, all samples with an AlScN top layer were coated with Mo to remove the electron charging effect during SEM measurement.

## 4. Experimental Results 

### 4.1. Single-Layer Film

Mo is an opaque substance, and Figure 4a depicts the waveform of a 0.1 μm thick Mo film. The acoustic wave is reflected from the Mo surface through the Mo/Si interface and back to the initial place at the wave peak position (dashed line 1). Mo layer thickness can be obtained by
(3)$$\begin{array}{c} T_{Mo}=\frac{t_{1}}{2}\times v_{Mo} \end{array}$$

Here, T is thickness of film, *t*_1_ is time at dashed line 1, v is sound velocity. Calculation of $T_{Mo}$ at (0,0) point is 99.8 nm. SEM result is 101.3 nm. The data consistency is 99%.

Al_0.9_Sc_0.1_N is a transparent material. The waveform of 0.6 μm Al_0.9_Sc_0.1_N is shown in Figure 4b. The waveform exhibits a typical Brillouin oscillation distribution. The difference in acoustic characteristics at the material interface is what causes the Brillouin cycle to change. The position of dashed line 1 in Figure 4b is where the Brillouin cycle changes, which means that the acoustic waves reach Al_0.9_Sc_0.1_N/Air interface at this time. Since the surface of the substrate Si absorbs the pump pulse. Acoustic waves are generated at the Si /Al_0.9_Sc_0.1_N interface. The Al_0.9_Sc_0.1_N layer thickness can be obtained by
(4)$$\begin{array}{c} T_{AlScN}=t_{1}\times v_{AlScN} \end{array}$$

Calculation of $T_{AlScN}$ at (0,0) point is 577 nm. SEM result is 574 nm. The consistency of data is also great.

### 4.2. Double-Layer Composite Film

As shown in Figure 5a, when the upper layer is transparent, the pulse is mainly absorbed by the opaque surface underneath the transparent layer, and acoustic wave is generated at the absorption interface. The subsequent process is the same as in a single transparent film, where the periodicity of the waveform changes dramatically at the interface between the transparent film and the air. Finding this transition point in the time domain (as indicated by dashed line 2 in Figure 5a) helps back-calculate the thickness of the upper transparent film by Equation (4). It is worth mentioning how to obtain the thickness of the lower opaque film from the waveform diagram. As shown in Figure 5a, it can be seen that at dashed line 1 there is also a small periodic fluctuation in the time domain plot. When the pump pulse is absorbed by the lower opaque film, the resulting sound wave propagates upward and downward at the same time. At the time indicated by dotted line 1, the acoustic wave that is transmitted round-trip in the opaque film will produce acousto-optical interference after being partially reflected at the substrate interface and partially entering the transparent layer. Therefore, we can get the thickness of the lower opaque layer by Equation (3).

Figure 5b shows the waveform of the upper layer with an opaque film. The opaque surface absorbs the pulse and generates the acoustic wave. Acoustic wave returns to the surface after a round-trip process in the Z-direction of the measured film, and changes refractive index (*n*) and extinction coefficient (*k*) in the region of the probe pulse radiation. Dashed line 1 calculates the thickness of Mo film by Equation (3). Compared with the dashed line 1, dashed line 2 has more round-trip process in the transparent layer. 

Therefore,
(5)$$\begin{array}{c} T_{AlScN}=\left(t_{2}-t_{1}\right)\times v_{AlScN} \end{array}$$

Sample E and Sample G are calculated as Mo (188 nm)/Al_0.9_Sc_0.1_N (1002 nm), Al_0.9_Sc_0.1_N (197 nm)/Mo (1025 nm). Both are very consistent with SEM data.

As mentioned above, picosecond ultrasound detection techniques focus only on the optical transparency of the material. Sample E differs from sample F only by the piezoelectric layer, and both are transparent materials for 515 nm wavelength. As shown in Figure 5c, the waveforms of sample E and sample F are very similar. The thickness can be likewise inferred from the position of the Brillouin cycle changing on the waveform. Hence, the picosecond ultrasonic detection technique has good generalizability. In addition, both sample waveforms appear as simple transparent material waveforms. This further demonstrates the sample’s tiny size and low density of anomalous grains.

### 4.3. Standard Piezoelectric Composite Structure

With the conclusion of the first two items, the waveform analysis of the composite sandwich structure film is straightforward. The thicknesses of the three layers can be deduced from the top to the bottom by the transition points of the wave peaks, as indicated with three dashed lines in Figure 6a. The calculation is Mo (208 nm) /Al_0.9_Sc_0.1_N (1000 nm)/Mo (194 nm), which also has great consistency with the SEM data.

However, it is worth noticing that the peaks at dashed line 2 are not the same as those at dashed lines 1,3. The reason for the difference is the difference in acoustic impedance at the three interfaces.

First, we need to understand the relationship between reflectivity change δR and strain η.
(6)$$\begin{array}{c} \delta R\left(t\right)=\int_{0}^{\infty }f\left(z\right)\eta \left(z,t\right)dz \end{array}$$

Here, *z* is distance from the surface, *t* is time domain plot. η (z,t) is a constant, and spatially exponentially decaying function near the surface. *f*(*z*) is the sensitivity function. *n*, *k* and *λ* all contribute to the shape and magnitude of the sensitivity function. This function decays exponentially and periodically [20]. The stress pulse changes due to the difference in acoustic impedance (*Z*) at the interface, Z is equal to the product of density and sound velocity [28]. As shown in Equation (7) and Equation (8), the strain transmission pulse is always positive. Therefore, we focus on observing the strain reflection pulse. As shown in Figure 6b, when *Z*_1_ > *Z*_2_, the strain reflection pulse is negative. From the acoustic impedance in previous literature [29,30], it is known that *Z*_Mo_ > *Z*_AlScN_ > *Z*_Si_. Therefore, the strain reflection pulse at interface Mo/AlScN (dashed line 1) and Mo/Si (dashed line 3) is negative and the strain reflection pulse at interface AlScN/Mo (dashed line 2) is positive. As shown by Equation (6), δR∝η. Therefore, the waveforms at dashed lines 1 and line 3 are close, and different from that at dashed line 2. This conclusion will be helpful for thinner and more multilayer film thickness probing.
(7)$$\begin{array}{c} Transmission:t_{21}=\frac{2Z_{1}\left(\frac{v_{1}}{v_{2}}\right)}{Z_{2}+Z_{1}} \end{array}$$
(8)$$\begin{array}{c} Reflection:r_{21}=\frac{Z_{2}-Z_{1}}{Z_{2}+Z_{1}} \end{array}$$

### 4.4. Application on 8-inch Wafers

The picosecond ultrasound measurement results and SEM data for all sample sites A–H are presented in Table 2. All data are at least 98% in agreement. Additionally, the picosecond ultrasound measurements have remarkable reproducibility. In this paper, 432 points on sample D were measured 10 times. The maximum average deviation is 2.1, while maximum deviation for a single point is 14.66.

After verifying the accuracy of picosecond ultrasonic detection by the above experiments, we will detect a piece of device wafer with a typical piezoelectric structure (substrate is Silicon-On-Insulator (SOI)). Picosecond ultrasound technology is noteworthy for its extremely fast measuring time in addition to its non-destructive and high accuracy benefits. A single-point simulation takes only about 8 seconds, and combined with a pulse emitter equipped with a high-precision robotic arm, we can obtain wafer-level thickness information in a short time. As shown in Figure 7, a thickness map with 432 points representing the thickness dispersion of the middle layer (AlScN) of piezoelectric stack on SOI substrate. The measurement data’s confidence is substantially increased by the wafer-level information, which also enables engineers to identify anomalous locations and more effectively solve process issues. For example, wafer-level data is very much needed in the field of nanoscale fabrication. [31,32].

## 5. Conclusions

In this paper, we conducted series of experiments to explain step-by-step the details of picosecond ultrasonic thin film thickness measurement technology. Taking AlScN stacking layers as the case study, the thickness measurements on both opaque and transparent thin films were examined. The noticeable difference on the single layer data plots was due to a typical Brillouin oscillation distribution. After completing the foreground studies on both single Moly and AlScN thin film layers, the composite structure thickness of two or three stacks was successfully extracted in 8-inch wafer scale. With the support of such a metrology tool, the wafer-level film thickness gradient was efficiently mapped to facilitate the in-line MEMS integration process. Benefiting from its advantage of sub-nm accuracy and measurement versatility with respect to all kinds of thin film materials, the picosecond ultrasonic thin film measurement technology can be highly adapted to the modern MEMS manufacturing industry.

## Figures and Tables

**Figure 1 micromachines-13-01916-f001:**
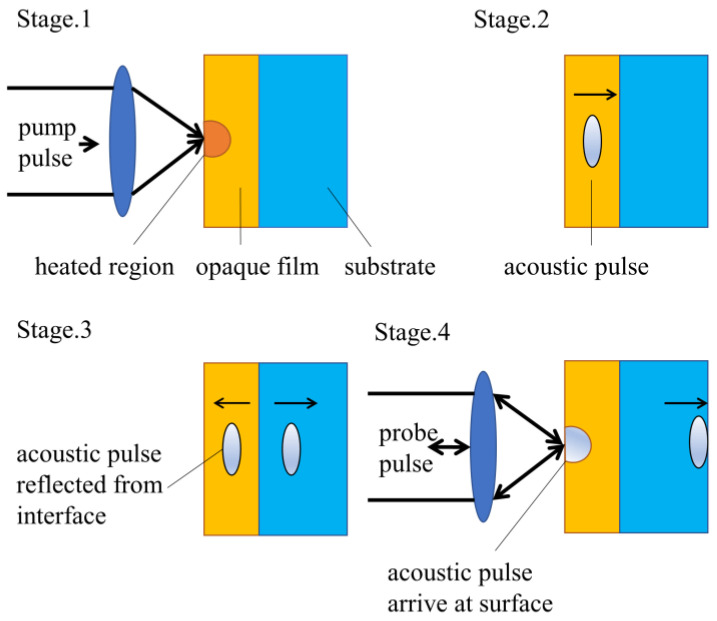
Sequence of events in opaque film.

**Figure 2 micromachines-13-01916-f002:**
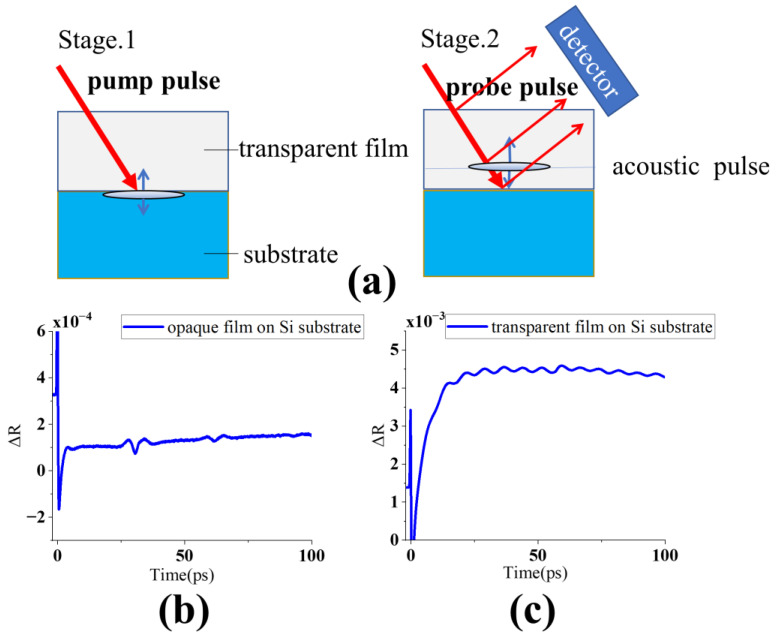
(**a**) Sequence of events in transparent film. (**b**) Typical reflectivity waveform for opaque film. ΔR= δR(t)/R. δR(t) is change amount of reflectivity. R is origin reflectivity. (**c**) Typical reflectivity waveform for transparent film on thermal background.

**Figure 3 micromachines-13-01916-f003:**
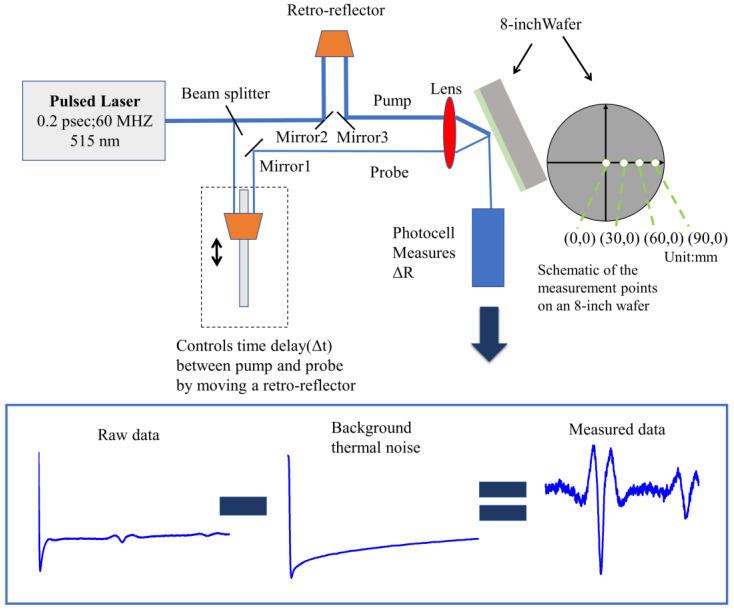
Schematic diagram of a picosecond laser pulse device and change of reflectivity after thermal background subtraction. Measurement points on wafer map is (0,0) (30,0) (60,0) (90,0). Unit is mm.

**Figure 4 micromachines-13-01916-f004:**
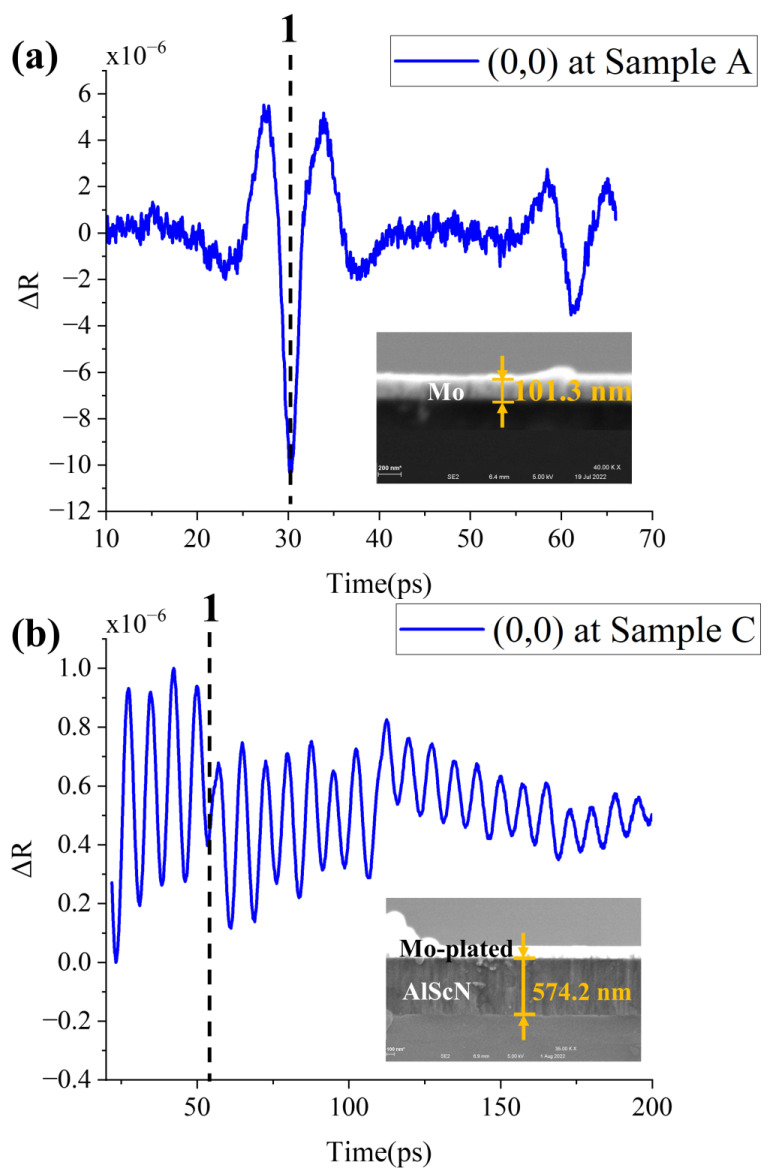
(**a**) Waveform diagram and SEM picture of sample A at (0,0) point. (**b**) Waveform diagram and SEM picture of sample C at (0,0) point. After picosecond ultrasound testing, the sample C is Mo-plated in order to reduce the influence of electron charging during SEM measurement.

**Figure 5 micromachines-13-01916-f005:**
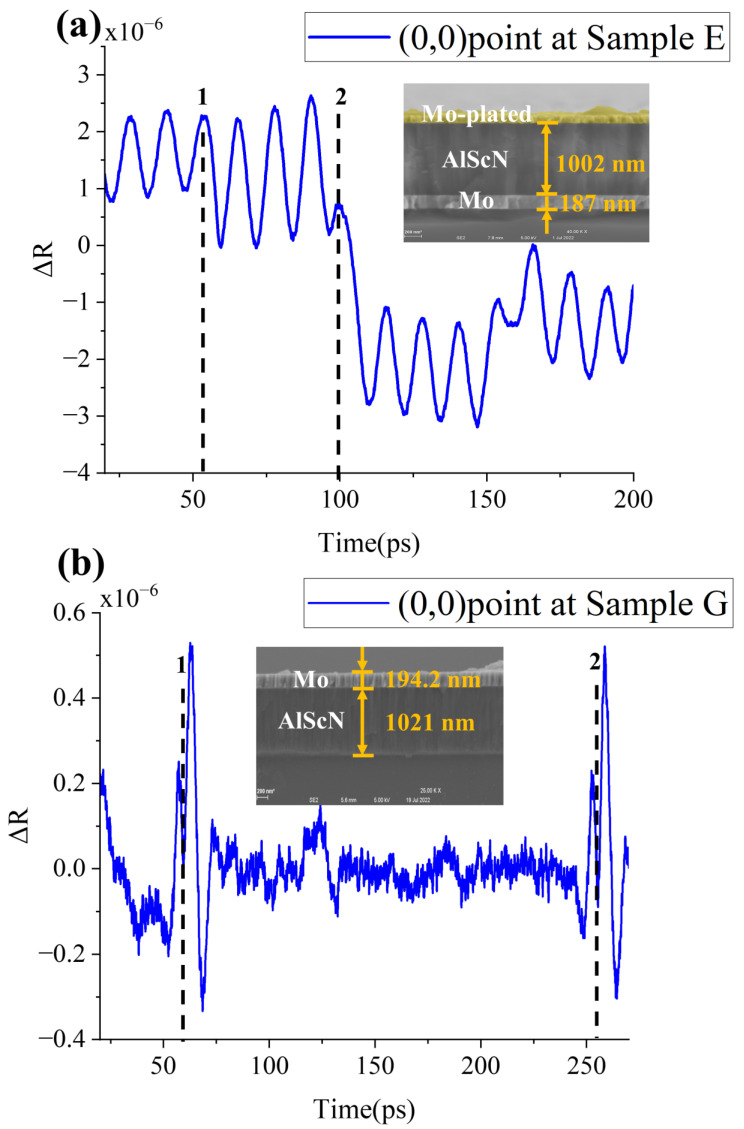
(**a**) Waveform diagram and SEM picture of sample E at (0,0) point. After picosecond ultrasound testing, the sample E is Mo-plated in order to reduce the influence of electron charging during SEM measurement. (**b**)Waveform diagram and SEM picture of sample G at (0,0) point. (**c**) Comparison of the waveforms of sample E and sample F at the (90,0) point.

**Figure 6 micromachines-13-01916-f006:**
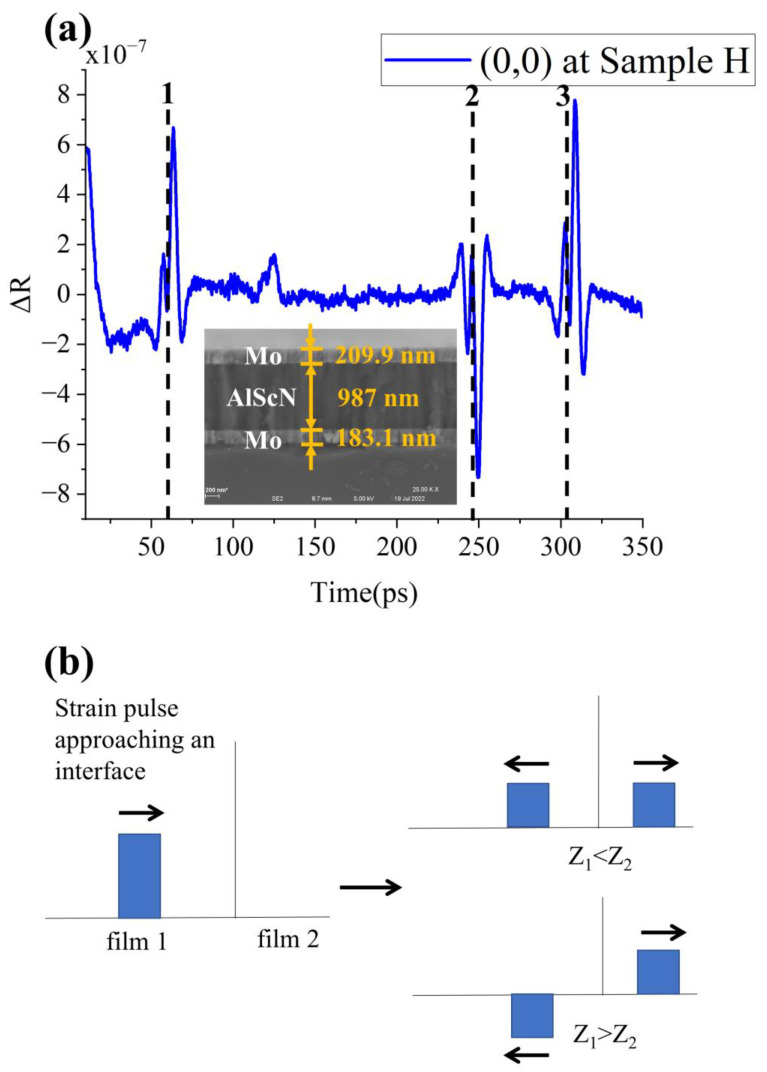
(**a**) Waveform diagram and SEM picture of sample G at (0,0) point. (**b**) Schematic of the strain reflection pulse.

**Figure 7 micromachines-13-01916-f007:**
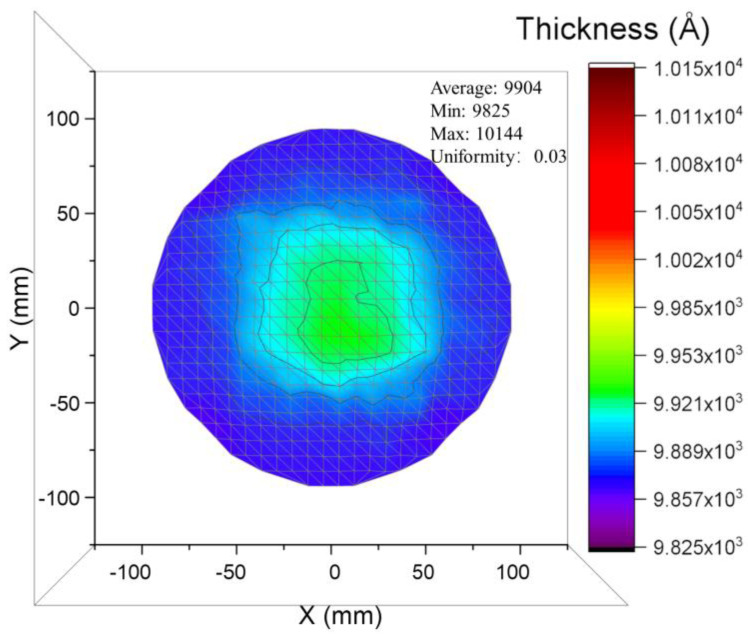
A thickness map with 432 points representing the thickness dispersion of the middle layer (AlScN) of device wafer.

**Table 1 micromachines-13-01916-t001:** Experimental sample information.

Sample	Structure
A	0.1 μm Mo
B	0.3 μm Mo
C	0.6 μm Al_0.9_Sc_0.1_N
D	1 μm Al_0.9_Sc_0.1_N
E	0.2 μm Mo + 1 μm Al_0.9_Sc_0.1_N
F	0.2 μm Mo + 1 μm Al_0.8_Sc_0.2_N
G	1 μm Al_0.9_Sc_0.1_N + 0.2 μm Mo
H	0.2 μm Mo + 1 μm Al_0.9_Sc_0.1_N + 0.2 μm Mo

**Table 2 micromachines-13-01916-t002:** Overview of picosecond ultrasound data and SEM data.

		(0,0) *	(30,0)	(60,0)	(90,0)	Consistency
A	picosecond ultrasound	998	1028	1009	976	99%
	SEM	1013	1005	1021	966	
B	picosecond ultrasound	2920	2962	2927	2838	99%
	SEM	2880	2941	2903	2854	
C	picosecond ultrasound	5777	5819	5582	5512	99%
	SEM	5742	5901	5678	5614	
D	picosecond ultrasound	10,006	9685	9699	9672	99%
	SEM	10,020	9666	9622	9634	
E	picosecond ultrasound	1884/10,024	1820/10,265	1730/9945	1628/9807	98%/98%
	SEM	1870/10,020	1786/10,080	1719/9769	1658/9881	
F	picosecond ultrasound	1940/10,143	1922/9989	1934/9874	1907/9807	98%/98%
	SEM	1926/10,103	1877/9881	1964/9639	1910/9742	
G	picosecond ultrasound	10,249/1965	9826/1997	9771/1978	9958/1906	98%/98%
	SEM	10,210/1942	9781/1786	9602/2010	9915/1876	
H	picosecond ultrasound	2077/10,003/1935	2114/9886/1985	1960/9760/1966	1905/9681/1911	98%/99%/98%
	SEM	2099/9870/1831	2188/9781/1965	1920/9691/1997	1954/9647/1876	

* (0,0) (30,0) (60,0) (90,0) are detection positions on sample (see Figure 3).

## Data Availability

The data are available within the article.

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
