# Peer review of "Nondestructive Wafer Level MEMS Piezoelectric Device Thickness Detection"

_micromachines, 2022, doi:10.3390/mi13111916_

Round 1
Reviewer 1 Report
Yongxin Zhou et all introduced a method for film thickness measurements. the results are good, however, some minor modifications are needed:
1. On Page 2, line 48, instead of a comma use a full stop after the reference [11]
2. On Page 3, line 103, instead of a comma use a full stop after Figure 3
3. What is the name and function of the orange tool that is placed on top? Please put its label (figure 3)
4. Figure 4b, please make sure of the scale of the y-axis
5. A comparison between this method and other methods is needed
Reviewer 2 Report
In this manuscript, the author demonstrated the picosecond ultrasonic pulse technique for measuring the thicknesses of piezoelectric thin films. The experiments and working principle of the method were described in detail. The experiment results show good agreement with the thicknesses of the film observed by SEM. However, it is unclear what the contribution of this manuscript is, since the authors simply present the data obtained from a commercially available equipment (the MetaPULSE G). In addition, the manuscript is unpolished with obvious typos. For instance, there are two sections 2.1. In line 97, there should be a space in "Figure 2". There should also be space before the unit. The writing style also needs to be improved to suit the academic writing style.
Reviewer 3 Report
The manuscript presents a nondestructive wafer scale thin film thickness measurement method by detecting the reflected picosecond ultrasonic wave transmitting between different interfacial layers. This topic is interesting for publication in Micromachines. However, the authors should revise the article according to the following, before I can make a positive recommendation regarding publication.
1. The authors should make further explanation of the measurement mechanism of opaque and transparent films. Why the optical energy is absorbed and converted to heat only in the opaque films?
2. The authors use a laser of 515 nm and 0.2 fs pulses for the measurement. Does this measurement method have special requirements for the wavelength and frequency of the pulsed laser?
3. In Fig. 3, thermal background noise is assumed to be the biggest disturbance to reflectivity waveform judgment, how to eliminate it?
4. In Fig. 4(b), it seems that a Brillouin cycle changes at about 120 ps, what is the reason?
5. In Fig. 5(a) and (b) the position of dashed line 1 are not at the crest, what is the reason? In Fig. 5(c), the waveforms of sample E and sample F at (90,0) point seems quite different, while the thickness of the films are same. How to eliminate the waveform variation caused by component differences?
6. The authors should give the uniformity value of the thickness through the map in Fig. 7.
7. Does this ultrasonic method could be used to measure other piezoelectric materials such as PZT film?
Round 2
Reviewer 2 Report
The authors provided additional information and explanations to elaborate on the method as well as highlight the novelty of their work. The manuscript can be accepted for publication after minor text editing. For example, in line 8, did the author mean "novel"?